# One-Step Reverse-Transcription Digital PCR for Reliable Quantification of Different Pepino Mosaic Virus Genotypes

**DOI:** 10.3390/plants9030326

**Published:** 2020-03-04

**Authors:** Nataša Mehle, Larisa Gregur, Alexandra Bogožalec Košir, David Dobnik

**Affiliations:** 1Department of Biotechnology and Systems Biology, National Institute of Biology, Večna pot 111, 1000 Ljubljana, Slovenia; larisa.gregur@goricko.info (L.G.); alexandra.bogozalec@nib.si (A.B.K.); david.dobnik@nib.si (D.D.); 2Goričko Nature Park Public Institute, Grad 191, 9264 Grad, Slovenia

**Keywords:** digital PCR, pepino mosaic virus, quantification, genotype specific

## Abstract

In recent years, pepino mosaic virus (PepMV) has rapidly evolved from an emerging virus to an endemic pathogen, as it causes significant loses to tomato crops worldwide. At present, the main control strategy for prevention of PepMV disease in tomato production remains based on strict hygiene measures. To prevent damage caused by PepMV, cross-protection is used in some countries. Reliable characterisation, detection and quantification of the pathogen are vital for disease control. At present, reverse-transcription real-time quantitative polymerase chain reaction (RT-qPCR) is generally used for this purpose. However, quantitative use of RT-qPCR is linked to standardised reference materials, which are not available for PepMV. In addition, many factors can influence RT-qPCR efficiencies and lead to lower accuracy of the quantification. In this study, well-characterised PepMV-genotype-specific RT-qPCR assays were transferred to two digital PCR (dPCR) platforms. dPCR-based assays allow absolute quantification without the need for standard curves, and due to the binary nature of the reaction, dPCR also overcomes many of the other drawbacks of RT-qPCR. We have shown that these newly developed and validated PepMV-genotype-specific dPCR assays are suitable candidates for higher-order methods for quantification of PepMV RNA, as they show lower measurement variability, with sensitivity and specificity comparable to RT-qPCR.

## 1. Introduction

Pepino mosaic virus (PepMV) belongs to the genus *Potexvirus* and has virions that are non-enveloped flexuous rods of 508 nm in length [1]. The PepMV genome is a 6410-nucleotide-long single strand of RNA [2]. It was first isolated in 1974 in Peru from pepino plants (*Solanum muricatum*) [1], and was reported as a pathogen of tomato (*Lycopersicon esculentum*) in 1999 in greenhouses in The Netherlands [3]. Shortly afterwards, PepMV became a major pathogen for tomato worldwide, with the first reports of its epidemic expansion in Europe, and later in North and South America [4]. PepMV can cause a wide range of symptoms that reduce the economic value of tomato crops. The symptoms typically include fruit marbling and leaf or stem necrosis [5,6,7]. The development and intensity of the symptoms are affected by various factors, including virus genotype, climate and tomato cultivar [6,8].

Pepino mosaic virus isolates show considerable genomic diversity, which appears to be linked to the geographic origins, specific hosts and/or the high mutation rates of RNA viruses. These differences appear to increase adaptation of PepMV to new environments [9]. There are five main PepMV genotypes currently recognized: Peruvian (including the original pepino isolate); European (Eur); American (US1, US2); Chilean (Ch2); and south Peruvian [10,11,12]. For several European countries and in North America, population genetic studies have revealed a shift in the dominant PepMV genotype from Eur to Ch2 [13,14]. Aggressive and highly virulent isolates that can cause severe symptoms and significant damage are distinguished from mild isolates by single nucleotide variations [8]. In addition, mixed infections of different PepMV genotypes have been reported [15,16,17]. Mixed infections can have greater detrimental effects on tomato crops due to symptom expression [15]. In mixed infections, there is also a high probability of recombination between the PepMV genotypes [15,16,17,18].

Pepino mosaic virus is highly infectious, and it is readily transmitted through mechanical means. It can be spread easily by plant-to-plant contact, and during routine cultivation procedures, while seed transmission might have a role in its long-distance spread [19]. Water has also been confirmed as a source of PepMV infection [20]. For the prevention of PepMV disease, there is the need for strict hygiene measures, which are currently the main control strategy for the protection of tomato production. However, severe outbreaks of PepMV continue in greenhouse tomato production in many European countries, and thus there is the need for further development of plant protection products. The first vaccine that was developed, PVM-01, contained a well-defined and characterised stable Ch2 mild isolate, and was registered in the European Union under Regulation 1107/2009 [21].

The PVM-01 vaccine was designed on the principle of cross-protection, which uses a mild isolate as a protective agent against more aggressive strains. Such cross-protection depends strongly on the genotype, and it is effective only under well-defined conditions [19]. For the vaccine, there is the need for accurate determination of the concentration of the PepMV. Therefore, reliable detection, quantification and genotype characterisation are the most important tools for disease control. In addition, accurate quantification of individual PepMV genotypes can be used for a variety of other applications, such as for evaluation of the resistance of plants to PepMV infection, of the variability of PepMV concentrations in different plant organs, and of the seasonal variations in virus concentrations, and also for estimation of the concentrations of different PepMV genotypes in a mixed infection.

Reverse-transcription real-time quantitative polymerase chain reaction (RT-qPCR) is at present still the method of choice for quantitative detection of the PepMV genotypes [22]. This quantification is based on standard curves, which are ideally prepared from certified reference material. As no reference material is available for PepMV, only relative quantification (i.e., between samples) is possible using RT-qPCR. Additionally, due to the high variability of the PepMV isolates, mismatches in primer or probe sequences can influence this quantification. Moreover, reverse transcription and PCR inhibitors can have major effects on PCR based assays.

In contrast to real-time quantitative PCR (qPCR), digital PCR (dPCR) can provide absolute quantification of target sequences without reliance on standard curves. By partitioning the samples into thousands of reactions before amplification, the copy number in the initial sample can be determined based on the number of positive and negative partitions after end-point PCR amplification, using Poisson’s statistics [23]. As well as eliminating the need for standard curves, dPCR has been shown to be more resilient to inhibitors [24,25]. In qPCR, a less-efficient amplification moves the cycle of quantification (Cq) to a higher value, while in dPCR, positive and negative partitions can be separated even with less efficient amplification. Thus, data from dPCR have lower sensitivity to variations in PCR amplification efficiencies. Furthermore, on the basis of this high-level sample partitioning, dPCR can provide very precise and accurate data even at particularly low target copy numbers [26].

Today there are several dPCR platforms available, which can be divided into two general groups: emulsion-based or droplet dPCR; and chip-based dPCR [27]. For the droplet dPCR platforms, the reaction mixture is divided into several individual droplets, while for chip-based dPCR, the reaction is divided into several chambers on a single plate or array. A combination of droplet-based and chip-based technology platforms is provided by the Naica system (Stilla Technologies) for crystal dPCR.

The aim of the present study was to evaluate two different dPCR platforms for quantification of PepMV RNA: QX100 or QX200 Droplet Digital PCR from Bio-Rad (henceforth as ‘QX100/200’); and Naica Crystal dPCR from Stilla Technologies (henceforth as ‘Naica’). We took the well-characterised assays for detection and genotyping of PepMV by qPCR, developed dPCR assays, and evaluated their performances. Despite some differences observed between these two dPCR platforms, which are indicated and discussed here, the newly developed reverse-transcription dPCR (RT-dPCR) assays could be potential candidate for higher order methods for absolute quantification of PepMV.

## 2. Results and Discussion

In this study, the following assays were transferred from RT-qPCR to RT-dPCR with the aim of absolute quantification without the need for standard curves: (i) for detection of all PepMV genotypes (henceforth as ‘PepMV-universal’); (ii) for detection of Peruvian and European PepMV genotypes (henceforth as ‘PepMV-Eur’); (iii) for detection of Ch2 and US2 PepMV genotypes (henceforth as ‘PepMV-Ch2’); and (iv) for detection of US1 PepMV genotype (henceforth as ‘PepMV-US1’). After the RT-dPCR assays were optimised for each dPCR platform used (i.e., QX100/200, Naica), their performances were validated and compared to RT-qPCR.

### 2.1. Optimisation of the Reverse-Transcription Digital Polymerase Chain Reaction (RT-dPCR) Assays on the QX100/200 Platform

The primers and probes for the RT-dPCR assays were the same as those used for the RT-qPCR assays. As the primer and probe concentrations and the annealing temperatures are known to influence the amplitude and clustering of the droplets, two concentrations of primers and probes were tested for the specific PepMV-Ch2, PepMV-Eur and PepMV-US1 assays, with three concentrations of primers and probe tested for the PepMV-universal assay. Additionally, different annealing–elongation temperatures were compared (from 55 °C to 65 °C). The droplet readouts demonstrated that the optimal resolution between positive and negative clusters, including the minimal amount of rain, was obtained with the same concentrations of primers and probe as used for RT-qPCR (Appendix A), with the optimal annealing–elongation temperatures of 55 °C to 57 °C (Appendix A). Therefore, an annealing–elongation temperature of 56 °C was used for the further studies.

### 2.2. Optimisation of the RT-dPCR Assays on the Naica Platform

The sequences and final concentration of primers and probes used for the Naica platform were the same as for the QX100/200. However, the fluorescent labels were changed for two of the assays: from 6-carboxyfluorescein (FAM) to hexachloro-fluorescein (HEX) for PepMV-Ch2; and from FAM to cyanine-5 (Cy5) for PepMV-US1. As the Naica platform has three detection channels, the change in fluorescent labels allowed the implementation of a triplex. At the time of this experimental work, Naica provided a low throughput of 12 reactions per run, compared to QX100/200 with 96 reactions per run. Multiplexing thus increased the throughput and improved the time efficiency.

A triplex assay that targeted the three different genotypes, of PepMV-Ch2, PepMV-Eur and PepMV-US1 was constructed, and its performance was compared to simplex assays. The RNA copy numbers of PepMV-Ch2 and PepMV-US1 quantified by triplex assay were significantly lower (34% and 93%, respectively) than those quantified by simplex assays. To explain this difference, assays were analysed for primer dimer formulation using the tool available from the National Institute of Standards and Technology (https://www-s.nist.gov/dnaAnalysis/primerToolsPage.do; [28]). Possible primer dimers were identified between the PepMV-US1 forward and PepMV-Ch2 reverse primers at 56 °C. Therefore, two duplex assays were designed and tested: (i) one that contained the PepMV-universal and PepMV-US1 primers and probes; and (ii) one that contained the PepMV-Eur and PepMV-Ch2 primers and probes. The coefficient of variation (CV) between three technical repeats was <8%, and the difference between the simplex and duplex assays was <8% (Table 1). As the differences between the copy numbers measured with the simplex and duplex assays were low, the duplex assays were chosen for further validation.

### 2.3. Validation of the RT-dPCR Assays

#### 2.3.1. Specificity, Repeatability and Reproducibility of the RT-dPCR Assays

The specificities of the RT-dPCR assays were confirmed through testing the RNA isolated from leaves of plants infected with different PepMV genotypes (i.e., Ch2, Eur, US1). All of these PepMV genotypes were detected with the PepMV-universal RT-dPCR (Appendix A). The genotyping in the RT-dPCR assays showed no cross-reactivities; i.e., the assays designed to discriminate between PepMV genotypes contained only negative clusters when the samples tested contained non-target PepMV genotypes, while the RNA samples of the target PepMV genotypes always produced clear separation of the positive cluster (Figure 1 and Figure 2).

The repeatability of the PepMV-genotype-specific RT-dPCR assays were evaluated through analysis of three replicates of RNA samples of different PepMV genotypes at various PepMV RNA concentrations. CVs <9% were obtained for the samples with >50 RNA copies/µL (Figure 3). As expected, higher CVs were obtained for the lowest concentrations of target RNA (Figure 3).

Assay reproducibility was assessed for six positive RNA samples tested in eight technical repeats on three separate days. The QX100/200 tests were also performed with different analysts and with different PCR cyclers. The mean copy numbers obtained here for the RNA samples of between 1616 and 16,896 PepMV RNA copies/µL for the different PepMV genotypes showed CVs <9%, while those with mean copy numbers between 14 and 152 PepMV RNA copies/µL showed CVs up to 22% (Table 2).

#### 2.3.2. Sensitivity and Linear Range for Different PepMV Genotypes by RT-dPCR Compared to Reverse-Transcription Real-Time Quantitative Polymerase Chain Reaction (RT-qPCR)

The sensitivities for the PepMV-universal and PepMV-genotype-specific RT-dPCR and the RT-qPCR assays were compared. The sensitivities for the RT-dPCR were a little higher for both the QX100/200 and the Naica platforms, compared to the RT-qPCR. Here, the RT-dPCR produced some signals (i.e., positive partitions) also at some dilutions that did not produce any signals for RT-qPCR (Table 3; PepMV-universal data are shown in Appendix A). Similar sensitivities as for dilution series of RNA prepared in RNAse-free water were observed for dilution series of RNA prepared in RNA isolated from leaves of healthy plants (Table 3). Further evaluation for the quantitative use of RT-dPCR was carried out only for the genotype-specific assays.

The correlation coefficients (*r*^2^) obtained by linear regression analysis showed a linear response (Figure 3; *r*^2^ > 0.99) of measured PepMV RNA copy numbers for all assays. Apart from the limitations presented by the assays themselves, the linear range of the RT-dPCR was limited by the number of partitions analysed. For the QX100/200 platform, the reaction mixture is divided into up to 20,000 droplets (mean of 13,792 droplets in this study), while the Naica system generated up to 35,000 droplets (mean of 25,220 droplets in this study). This means that the Naica system allowed quantification of higher copy number concentrations than the QX100/200 system. However, for the highest concentrations of PepMV-US1 tested (i.e., 10^−3^ dilution), the numbers of PepMV copies were in the upper range of both instruments for quantification, at near saturation, where 100% (or ~100%) of the droplets analysed contained PepMV copies (see Appendix A). At these concentrations, although PepMV was detected, there was a relatively high probability of error [29]. In this situation, for higher PepMV loads, the RNA samples need to be diluted before being measured.

The quantifications by RT-dPCR and RT-qPCR of the RNA samples with, approximately, from 1000 to 50,000 PepMV RNA copies/µL resulted in CVs between technical replicates of 1% to 5% for QX100/200, 1% to 8% for Naica, and 3% to 15% for RT-qPCR (Figure 3). For quantification of the RNA samples with approximately from 50 to 1000 PepMV RNA copies/µL, the CVs between the technical replicates were from 3% to 7% for QX100/200, 4% to 8% for Naica, and 6% to 21% for RT-qPCR (Figure 3). As expected, the quantification of the RNA samples with, approximately, from 1 to 50 PepMV RNA copies/µL gave higher CVs (QX100/200, 14–26%; Naica, 19–43%; RT-qPCR, 33–51%) (Figure 3). These data indicate the lowest overall measurement variability for QX100/200, followed by Naica, with the overall measurement variability of RT-qPCR as the highest.

A further important advantage of RT-dPCR is that an absolute RNA copy number is obtained for the samples, which is not possible for RT-qPCR. Therefore, as no reference material was available for PepMV, RT-qPCR only provided relative quantification. However, strategies are available for indirect copy-number determination in a sample, such as the use of limiting dilutions, as discussed in our previous study on PVY, for example [30]. In the present study, the data from QX100/200 was used to calibrate the RT-qPCR and, therefore, as might be expected, the quantification according to these two PCR systems was in agreement (Figure 3). However, there were differences for the RT-dPCR quantification (Table 3, Figure 3). The actual copies quantified by Naica were higher than those quantified by QX100/200, by 37.5% on average (for samples with 50 to 50,000 PepMV RNA copies). It has been shown that accurate droplet volume and its variability is critical for the accuracy of any absolute quantification with dPCR [31,32,33,34]. Bogožalec Košir et al. [33] showed that the droplet volume of the QX100/200 platform depends mainly on the droplet generator and the supermix used. No similar studies regarding droplet volumes for the Naica platform have been carried out to date. Here, the droplet volumes assigned by the manufacturer were used for calculation of the copy numbers of PepMV RNA (QX100/200, 0.85 nL; Naica, 0.43 nL). However, using the 0.739 nL droplet volume for the QX100/200 copy number that was measured by Bogožalec Košir et al. [33] for the DG32 automated droplet generator (as used in the present study), the discrepancies between these assays were lower (19.5% on average; for samples with 50 to 50,000 PepMV RNA copies; data not shown). In the study of Bogožalec Košir et al. [33], the droplet volume determined for dPCR was used for quantification of the DNA target and, therefore, the supermix they used was also different from that used in the present study. Discrepancies between copy numbers of PepMV RNA determined using dPCR indicate that if precise quantification is required, there is the need for the droplet volumes to be assessed for both platforms, using the relevant type of supermix.

## 3. Materials and Methods

### 3.1. Samples and Sample Preparation

The PepMV genotypes used in this study were Eur, Ch2 and US1. PepMV-Eur (isolate 99901066) was kindly provided by Dr J.Th.J. Verhoeven (NVWA, Netherlands), and PepMV-Ch2 (isolate 1906) and PepMV-US1 (isolate ST 08/008) were kindly provided by Dr Inge Hanssen (Scientia Terrae Research Institute, Belgium). PepMV isolates were propagated on *Lycopersicon esculentum* cv. ‘Moneymaker’.

Total RNA extraction was carried out for infected and healthy plant leaf material with RNeasy Plant mini kits (Qiagen, Hilden, Germany), according to the manufacturer recommendations, with some minor modifications. Briefly, 200 mg plant material was homogenised in 900 µL lysis buffer (RLT buffer without mercaptoethanol; Qiagen), then 600 µL of homogenate was loaded onto RNeasy Mini Spin columns. RNA elution from these RNeasy Mini Spin columns was carried out as two consecutive applications of 50 µL (i.e., 100 µL total) of RNase-free warm water (65 °C). Buffer controls were included with all isolations (i.e., for negative isolation control), to monitor potential contamination through the extraction procedures.

### 3.2. Assays Used

The RNA of PepMV was amplified using the PepMV-universal primers and probe described by Ling et al. [16], and by three PepMV-genotype-specific RT-qPCR assays described by Gutierrez-Aguirre et al. [22]: (i) Eur-rep (assay for specific detection of Eur and Peruvian genotypes; PepMV-Eur RT-qPCR); (ii) Ch2 and US2-Cp (assay for specific detection of Ch2 and US2 genotype; PepMV-Ch2 RT-qPCR); and (iii) US1-Cp (assay for specific detection of US1 genotype; PepMV-US1 RT-qPCR). All of the probes were labelled with FAM as the reporter and carboxy tetramethylrhodamine (TAMRA) as the quencher. There were exceptions for the probes used for the Naica platform, where double-quenched ZEN probes (Integrated DNA Technologies) were used. The probes were labelled as follows: (i) probe for PepMV-universal and specific probe for PepMV-Eur with FAM-ZEN; (ii) specific probe for PepMV-Ch2 with HEX-ZEN; (iii) specific probe for PepMV-US1 with Cy5-ZEN.

### 3.3. RT-qPCR

The RT-qPCR assays were carried out with AgPath-ID One-Step RT-qPCR mix (Ambion, USA). The final reaction volume (10 µL) included 2 µL RNA sample, 900 nM primers and 200 nM probe for the specific PepMV-Ch2, PepMV-Eur and PepMV-US1 assays, or 200 nM primers and 400 nM probe for the PepMV-universal assay. Each RT-qPCR run included a non-template control and a negative isolation control. RT-qPCR was performed with 384-well plates (Applied Biosystems, Foster City, CA, USA), with reactions run as at least two technical repeats on an ABI Prism 7900HT Fast Detection system (Applied Biosystems, Foster City, CA, USA). The condition for the cycling were 10 min at 48 °C, 10 min at 95 °C, and 45 cycles of 15 s at 95 °C, with 1 min at 60 °C. Reactions were considered positive if an exponential amplification curve was produced that could be distinguished from the negative controls; with no production of an exponential amplification curve, the reaction was considered negative. Fluorescence acquisition and of Cq determination was carried out with the SDS 2.4 software (Applied Biosystems, Foster City, CA, USA). Automatic setting of the baseline was used, with manual setting of the fluorescence threshold at 0.3. This represented a level above the baseline but low enough to remain within the exponential increase region of the amplification curve.

### 3.4. Optimisation of RT-dPCR: QX100/200

The QX100/200 RT-dPCR assays were carried out with One-Step RT-ddPCR Advanced kits for probes (Bio-Rad, CA, USA). The samples were each analysed as at least two technical repeats, with each run including a non-template control. The final volume of the reaction mixtures was 20 µL, which included 5 µL Supermix (Bio-Rad, CA, USA), 2 µL reverse transcriptase (final concentration, 20 U/µL; Bio-Rad, CA, USA), 1 µL 300 mM dithiothreitol (Bio-Rad, CA, USA), forward and reverse primers, and the probe, with molecular grade RNAse-free water, and finally 4 µL RNA sample. Different final concentrations of primers and probe were compared to optimise the RT-dPCR reactions: (i) the same as for the RT-qPCR assays described above; (ii) 900 nM primers (specific PepMV-Ch2, PepMV-Eur, PepMV-US1 assays)/450 nM primers (PepMV-universal assay) and 250 nM probe; (iii) 300 nM primers and 100 nM probe (PepMV-universal assay).

Preparation of the reaction mixtures was carried out in 96-well polypropylene plates (Eppendorf, Hamburg, Germany). These were then heat-sealed using pierceable foil, and the droplets were generated with an automated droplet generator (DG32; Bio-Rad, CA, USA) using automated droplet generation oil for probes (Bio-Rad, CA, USA) and a cartridge for automated generation of the droplets (DG32; Bio-Rad, CA, USA). Droplets suspensions were collected in separate 96-well polypropylene plates. After the heat sealing of these plates with droplets with pierceable foil, they were transferred to a thermal cycler (T100 or C1000 Touch; Bio-Rad, CA, USA). The conditions for the thermal cycling included 60 min reverse transcription at 50 °C and 10 min enzyme activation at 95 °C, which were followed by 40 cycles using a two-step thermal profile, of 30 s denaturation at 95 °C and 1 min annealing–elongation. Finally, this was followed by 10 min enzyme deactivation at 98 °C. For best possible separation of clusters of positive and negative droplets, annealing–elongation temperatures were tested between 55 °C and 65 °C. The ramp rate was set as 2.5 °C/s. Following the thermal cycling, plates were transferred into a droplet reader (QX100 or QX200; Bio-Rad, CA, USA). A fluorescence amplitude threshold was used to discriminate the positive droplets containing the amplification products from the negative droplets, using the QuantaSoft software, version 1.7.4 (Bio-Rad, CA, USA). The manual setting of the threshold was to just above the negative droplet cluster, as visualised using the fluorescence amplitude versus event number in the FAM channel (Figure 1). Samples with up to three positive droplets with fluorescence intensities that were very low (i.e., <1000 above the threshold) were considered negative. Rejection from subsequent analysis of the data generated by the QuantaSoft software was based on analysis of <9000 droplets per 20 µL reaction (i.e., <7.65 µL effective reaction size), or also when there were >99.99% positive droplets (which indicated reaction saturation) (Appendix A).

### 3.5. Optimisation of RT-dPCR: Naica

The Naica RT-dPCR assays were carried out with qScript XLT One-Step RT-qPCR ToughMix (Quanta BioSciences, Beverly, MA, USA). Each sample was analysed as at least two technical repeats, with inclusion of a non-template control for each run. The final volume of the reaction mixtures was 25 µL, which included 12.5 µL ToughMix, 2.5 µL fluorescein (FITC; final concentration, 100 nM), forward and reverse primers, and the probe, with molecular grade RNAse-free water, and 4 µL RNA sample. Simplex, duplex (PepMV-universal + specific PepMV-US1, and specific PepMV-Eur + PepMV-Ch2) and triplex (PepMV-Ch2 + PepMV-Eur + PepMV-US1) reactions were tested and compared. The final concentration of each primers and each probes used for the Naica platform in simplex, duplex and triplex reactions were the same as for the RT-qPCR assays described above.

The reaction mixtures were loaded into 4-well Naica sapphire chips (Stilla Technologies, Villejuif, France), and the chips were transferred into the Naica Geode (Stilla Technologies, Villejuif, France) for droplet generation and PCR cycling. The droplets were generated at 40 °C, followed by thermal cycling of 10 min reverse transcription at 50 °C, 1 min enzyme activation at 95 °C, then 45 cycles of a two-step thermal profile with 30 s denaturation at 95 °C, and the 15 s annealing–elongation at 56 °C. Following thermal cycling, the chips were scanned using a Crystal reader (Stilla Technologies, Villejuif, France) for the following exposure times: (i) 70 ms for FAM (blue channel); (ii) 125 ms for HEX (green channel); (iii) 25 ms for Cy5 (red channel). Data analysis used the Cristal Miner software, version 1.4.3 (Stilla Technologies, Villejuif, France). After application of spillover compensation to the raw fluorescence data [35], a threshold was applied to discriminate the positive droplets from the negative droplets, using the automated tool supplied by the software (Figure 2). Samples with up to three positive droplets with fluorescence intensities that was very low (i.e., <1000 above the threshold) were considered negative. Rejection from subsequent analysis of the data generated by the Cristal Miner software was based on analysis of <15,000 droplets per 25 µL reaction (i.e., <6.45 µL effective reaction size), or also when there were >99.99% positive droplets (which indicated reaction saturation).

### 3.6. Validation of RT-dPCR Assays for Quantification of Different PepMV Genotypes

The optimised RT-dPCR assays underwent validation based on the recommendations of the European and Mediterranean Plant Protection Organisation [36]. The RT-dPCR assay specificities were determined experimentally through two (QX100/200) or three (Naica) technical replicate measurements of RNA isolated from leaf material infected with PepMV-Ch2, PepMV-Eur and PepMV-US1. To determine the reproducibility, two dilutions of each RNA sample, one with medium and the other with low PepMV RNA concentrations were prepared and stored below −15 °C (Table 2). Samples without PepMV RNA were used as negative controls. Three runs were carried out for each experiment (on three different days), with each run including sample measurements as two or three technical repeats. The technical repeats were carried out under repeatability conditions within each single run: the analyst, reaction mix and instruments were the same, and the chemicals and cartridges were from the same batch. At least one additional condition was changed between the QX100/200 runs, in order to contrast: two analysts, two thermal cyclers for amplification (T100, C1000 Touch; Bio-Rad, CA, USA) and two droplet readers (QX200, QX100; Bio-Rad, CA, USA).

The sensitivities and linear ranges of the PepMV-genotype-specific RT-dPCR assays were tested on a dilution series (10-fold) of PepMV-positive RNA samples (in RNAse-free water), over the RNA dilution range of 10^−3^ to 10^−8^. RNA was extracted from test plants infected with PepMV-Ch2, PepMV-Eur and PepMV-US1. The results were compared to the RT-qPCR results. For PepMV-universal assays, only the sensitivity was tested, using a dilution series (10-fold; from 10^−3^ to 10^−8^) of a mix of PepMV-Ch2, PepMV-Eur and PepMV-US1 RNA-positive samples (the mix contained the same volume of RNA of each PepMV genotype). Additionally, a dilution series (10-fold; from 10^−4^ to 10^−9^) of PepMV-Ch2 in RNA isolated from leaf material of healthy tomato plants was prepared and tested. Each dilution was analysed as three technical repeats for each assay. The same RNA dilutions were used for each assay.

### 3.7. Data Analysis

Relative quantification was carried out using the standard curve method [37]. For comparisons of output data from the RT-qPCR and RT-dPCR, Cq was used to define the RNA copy numbers. For this conversion, actual copy numbers were initially determined with RT-dPCR QX100/200, as the calibrator [30]. Cq conversion was defined by the most accurately calculated PepMV RNA concentration from RT-dPCR (i.e., which showed the lowest CV across technical repetitions). This PepMV RNA concentration was then used for the back calculation of those in the other dilutions. Using linear regression, the equation for calculation of the number of detected targets using the Cq of the samples was defined, which was based on the copy numbers determined [38].

## 4. Conclusions

The aim of the present study was for the development of an accurate method to provide absolute quantification of different PepMV genotypes without the need for standard curves. This was achieved by the transfer of PepMV-genotype-specific RT-qPCR assays to two dPCR platforms. In comparison with RT-qPCR, the PepMV-genotype-specific RT-dPCR assays showed sensitivities that were similar or were up to 10-fold greater, with better overall repeatability, for both of the tested platforms. With the aim of designing an assay that can detect all PepMV genotypes by dPCR, the PepMV-universal RT-qPCR assay also underwent transfer to the two RT-dPCR platforms. The PepMV-universal assay showed similar performance in dPCR and qPCR; however, the validation of the PepMV-universal assay was only carried out for the purpose of PepMV detection, and not for quantification.

The high accuracy of this newly developed PepMV-genotype-specific RT-dPCR will provide for greater accuracy for PepMV RNA quantification in studies that are needed to support disease control, and for more precise quantification of PepMV in vaccines used for cross-protection. Moreover, the RT-dPCR assays can also be used for characterisation of the calibrators needed for standard curves in RT-qPCR. However, to more accurately determine the copy numbers, further studies are necessary to determine a more accurate droplet volume for the One-Step RT-ddPCR Advanced kits for probes used in combination with the QX100/200 platform, and to determine a more accurate droplet volume for the Naica system. The general unavailability of standardised reference materials represents a problem for detection and quantification of plant pathogens, and so the present study represents the first step in the development of calibrators or higher order quantitative methods for application to not only PepMV, but also to further plant viruses and further plant pathogens.

## Figures and Tables

**Figure 1 plants-09-00326-f001:**
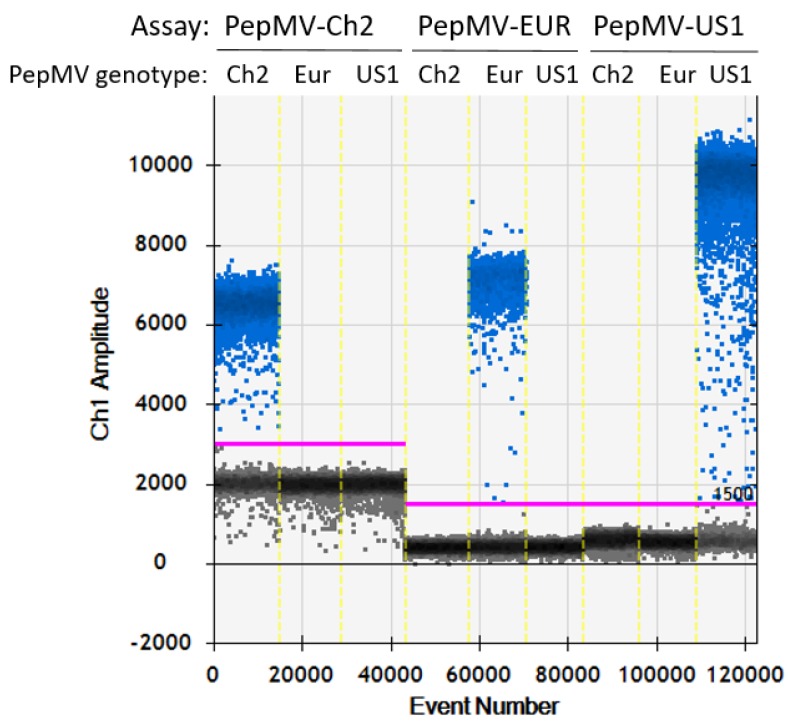
Specificity of the RT-dPCR assays tested on the QX100/200 platform. Specificity was confirmed by testing different PepMV genotyping assays on samples that contained different PepMV genotypes. RNA isolated from leaves of infected test plants with PepMV-Ch2 (dilution, 10^−4^), PepMV-Eur (dilution, 10^−4^) and PepMV-US1 (dilution, 10^−5^) were tested. For each sample, droplets are depicted according to the event (number of droplet as read during reading; x axis) and its fluorescence (Ch1 Amplitude; y axis). The setting of the threshold for droplet positivity was done manually, and is defined by the thick pink horizontal line (left, 3000; right, 1500). Each sample was tested as two technical repeats. As the results did not differ between the technical repeats, the representative results of one of the technical repeats are shown.

**Figure 2 plants-09-00326-f002:**
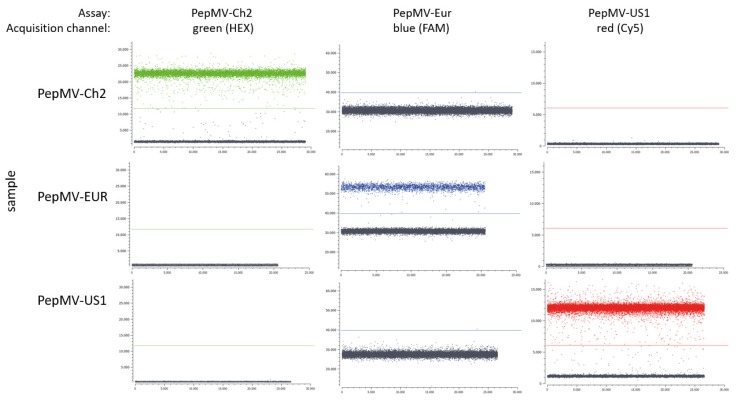
Specificity of RT-dPCR assays tested on the Naica platform. Specificity was confirmed by testing different PepMV genotyping assays on samples containing different PepMV genotypes. RNA isolated from leaves of infected test plants with PepMV-Ch2 (dilution, 10^−4^), PepMV-Eur (dilution, 10^−4^) and PepMV-US1 (dilution, 10^−5^) were tested. For each sample, the plot representing droplet fluorescence intensity in the green/blue/red acquisition channel on the y axis, and droplet index (a number assigned to each droplet during analysis) on the x axis. The setting of the threshold for the droplet positivity was done automatically by the Crystal Miner software, and is shown by the horizontal lines. As the results did not differ between the technical repeats, the representative results of one of the technical repeats are shown.

**Figure 3 plants-09-00326-f003:**
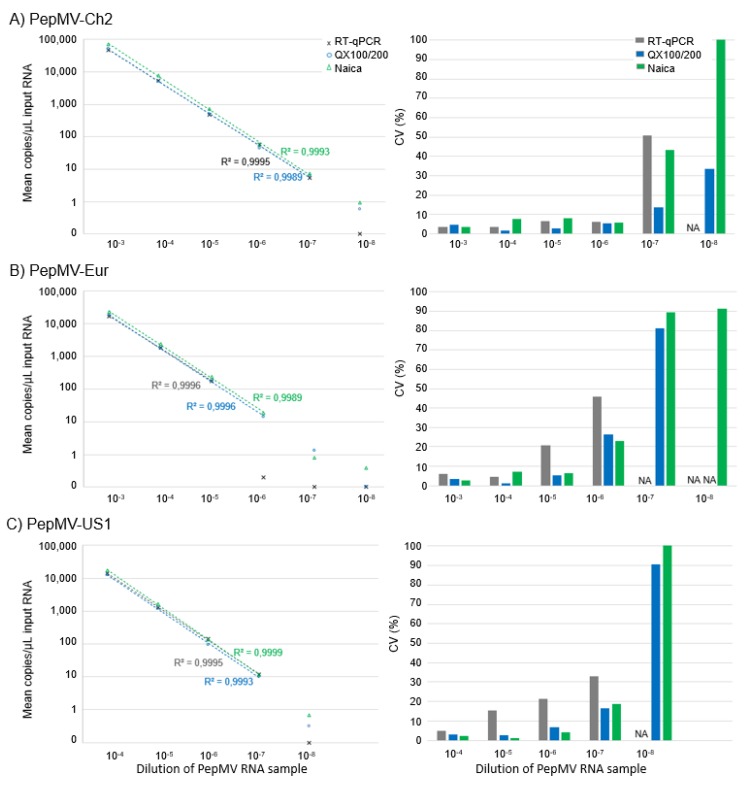
Parallel PepMV-genotype-specific RT-dPCR (blue, QX100/200; green, Naica) and RT-qPCR (grey) analyses of the different genotypes of PepMV RNA. Left: correlations between serial dilutions and concentrations of PepMV RNA determined (*n* = 3, for each dilution and each method). Linear regression is plotted for each genotype tested with each method, with the correlation coefficients (*r*^2^) given. Right: precision of the assays, as CV (%) of the PepMV RNA copy numbers measured. NA, not applicable (below the limit of detection).

**Table 1 plants-09-00326-t001:** Comparison of pepino mosaic virus (PepMV) RNA quantification using simplex and duplex reverse-transcription digital polymerase chain reaction (RT-dPCR) assays performed on Naica.

Genotype	Simplex		Duplex		Simplex vs. Duplex	
	Input RNA (mean; copies/µL)	CV (%)	Input RNA (mean; copies/µL)	CV (%)	Input RNA (Δ; copies/µL)	Δ Duplex (%Simplex)
Mix ^a^	20,573	1.1	22,144	3.8	1571	7.6
Ch2	7021	1.4	7292	7.9	271	3.9
Eur	2398	3.1	2253	7.1	145	6.0
US1	16,717	1.6	17,027	2.3	310	1.9

^a^ mix of PepMV-Ch2, PepMV-Eur and PepMV-US1 RNA-positive samples (the mix contained the same volume of RNA of each PepMV genotype) analysed with PepMV-universal RT-dPCR on Naica; CV, coefficient of variation.

**Table 2 plants-09-00326-t002:** Reproducibility of the RT-dPCR assays for PepMV RNA.

Sample	Runs (n)	Replicates (n)	QX100/200		Naica	
			Input RNA (mean; copies/µL)	CV (%)	Input RNA (mean; copies/µL)	CV (%)
PepMV-Ch2-M	3	8	5483	7.9	7552	5.6
PepMV-Eur-M	3	8	1616	8.2	2165	6.1
PepMV-US1-M	3	8	12,905	4.1	16,896	3.0
PepMV-Ch2-L	3	8	51	14.3	71	20.0
PepMV-Eur-L	3	7–8 ^a^	14	21.2	21	21.8
PepMV-US1-L	3	6–8 ^b^	110	13.9	152	15.1
PepMV-Ch2-zero	3	8	0	NA	0 ^c^	NA
PepMV-Eur-zero	3	5	0	NA	0 ^c^	NA
PepMV-US1-zero	3	5	0	NA	0	NA

-M, medium; -L, low; concentrations of PepMV RNA; -zero, no PepMV RNA (negative controls); NA, not applicable; CV, coefficient of variation; ^a^ 7 replicates tested with QX100/200 and 8 replicates tested with Naica; ^b^ 8 replicates tested with QX100/200 and 6 replicates tested with Naica; ^c^ In a few replicates, up to three positive droplets were observed with very low fluorescence intensity (but above the threshold).

**Table 3 plants-09-00326-t003:** Sensitivities of the RT-qPCR and RT-dPCR assays for PepMV.

Sample	RNA Dilution	RT-qPCR Cq	Input RNA (copies/µL)		
			RT-qPCR ^a^	QX100/200 ^b^	Naica ^b^
PepMV-Ch2	10^−3^	21.1 ± 0.1	46,737 ± 1648	51,907 ± 2337 ^e^	71,818 ± 2502
	10^−4^	24.4 ± 0.1	5350 ± 186	5031 ± 90	7495 ± 588
	10^−5^	28.2 ± 0.1	465 ± 30	500 ± 14	727 ± 58
	10^−6^	31.5 ± 0.1	58 ± 3.5	44 ± 2.4	57 ± 3.4
	10^−7^	35.4 ± 1.0	5 ± 2.7	6 ± 0.9	7 ± 3.1
	10^−8^	Negative ^c^	0 ^c^	1 ± 0.2	1 ± 1.4 ^e^
PepMV-Eur	10^−3^	22.4 ± 0.1	16,485 ± 1005	17,254 ± 603	21,777 ± 610
	10^−4^	25.6 ± 0.1	1822 ± 86	1713 ± 23	2315 ± 164
	10^−5^	29.1 ± 0.3	170 ± 35	175 ± 9	235 ± 15
	10^−6^	38.7 ± 0.8	0 ± 0.1	14 ± 3.8	18 ± 4.2
	10^−7^	Negative	0	1 ± 1.1	1 ± 0.7 ^d^
	10^−8^	Negative	0	0 ^c^	0 ± 0.4 ^d^
PepMV-US1	10^−3^	20.5 ± 0.1	109,074 ± 8848	NA	115,505 ± 19,153 ^e^
	10^−4^	23.8 ± 0.1	13,888 ± 676	12,495 ± 384	17,500 ± 396
	10^−5^	27.6 ± 0.2	1288 ± 196	1215 ± 33	1616 ± 20
	10^−6^	31.1 ± 0.4	142 ± 30	94 ± 6.4	135 ± 6
	10^−7^	35.1 ± 0.6	12 ± 3.9	10 ± 1.7	12 ± 2.3
	10^−8^	Negative	0	0 ± 0.3^d^	1 ± 0.8 ^d^
PepMV-Ch2 (in RNA from healthy plants)	10^−4^	25.1 ± 0.1	4297 ± 184	4608 ± 175	7273 ± 127
	10^−5^	28.4 ± 0.1	440 ± 34	459 ± 6	716 ± 15 ^e^
	10^−6^	31.8 ± 0.2	45 ± 5.0	44 ± 5.8	70 ± 2.2
	10^−7^	34.8 ± 0.4	6 ± 1.6	5 ± 1.2	9 ± 0.8 ^e^
	10^−8^	37.8 ± 0.6 ^d^	1 ± 0.4 ^d^	0	1 ± 0.9
	10^−9^	Negative	0	0 ± 0.4 ^d^	1 ± 0.5 ^d^

Data are means ± standard deviation (*n* = 3); Cq, quantification cycle; NA, not applicable (as most droplets analysed contained target copies; therefore it was not possible to apply Poison’s law, and so the concentration of the targets could not be determined); ^a^ RT-dPCR measurements of 10^−4^ were used as calibrators to calculate PepMV concentrations in the dilutions of the samples. Linear regression was then applied to obtain the equation for calculation of the number of targets detected from the Cq of the samples; ^b^ Selected parameters from the RT-dPCR analysis are listed in Appendix A; ^c^ Positive only one of three replicates with signal Cq >37 (RT-qPCR)/only one positive droplet in RT-dPCR; ^d^ Signal was observed in two of three replicates; ^e^ concentration of targets determined for only two replicates because in one replicate almost all droplets analysed contained target copies, or because <9000/<15,000 droplets were analysed per 20 µL (QX100/200)/25 µL reaction (Naica), or due to a pipetting error.

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
