# Peer review of "One-Step Reverse-Transcription Digital PCR for Reliable Quantification of Different Pepino Mosaic Virus Genotypes"

_plants, 2020, doi:10.3390/plants9030326_

Round 1
Reviewer 1 Report
In this work, the authors describe specific and very sensitive assays for the absolute quantification of different PepMV isolates by RT-dPCR. These assays do not need the use of a standard curve for quantification and have less variation between replicates compared to the RT-qPCR assays, among the other advantages mentioned above. In my opinion, the experimental design is appropriate for the main purpose of this work, which is to develop a precise method for the absolute quantification of viral isolates for which no reference material is available. The performance of the experiments presented here is also correct and I only have some curiosities or minor suggestions to recommend the publication of this work.
I wonder if it is possible to use known concentrations of viral RNA from disassembled virions instead of total RNA from infected tomato plants, at least to verify the sensitivity and linear range for different PepMV isolates using RT-dPCR or RT-qPCR. In my opinion, the results would be more accurate using viral RNA. Viral RNA could be diluted in preparations of total RNA from uninfected plants instead of in pure water. In addition, a standard curve of serial dilutions of viral RNA could be used for absolute quantification in the RT-qPCR for the experiment of Table 3 instead of using the equation performed with the RT-dPCR experiment.
For a better understanding of the figures, I believe that a description of what is represented on both x and y axes should be added to the legend of at least figures 1 and 2.
Review paragraph from line 309 to line 315. The explanation of how to prepare the reaction is not clear.
The concentration of the probe and primers used in the optimization of the Naica RT-dPCR are not provided (around line 334).
Around line 358, it is said that “three aliquots of each RNA sample with medium, low and zero (negative control) concentration of PepMV were prepared”. In my opinion, the term “aliquots” is not used correctly here, as it seems to refer to three different aliquots with different concentration of the same PepMV infected sample, although at least the “zero” sample should be a healthy sample.
Author Response
Dear Reviewer 1,
We would like to thank you for your valuable comments, which helped to improve the manuscript. We have carefully revised the manuscript based on your and other suggestions. Please see below a response to the comments provided by you.
- We agree that using viral RNA would be much better. But to prepare viral RNA would be very expensive and time consuming. According to our experiences this would take months and it happens several times that it fails. Therefore, this was not possible to be done in the frame of that project. As a consequence, in reality for most of applications (if not in all) in the field of plant virology, the concentration will be measured on total RNA.
- As suggested, for a better understanding of Figure 1 and 2, descriptions about what is represented on x and y axes are added to the legend.
- The paragraph where we explain how the reaction mix was prepared is now slightly modified and hopefully it is now more clear (see lines 329-337)
- The concentration of the probes and primers used in Naica RT-dPCR is now provided (see lines 372-374)
- Yes, the term aliquots is not correct, and zero sample is actually healthy sample (negative control). This is corrected now (see lines 399-401)
Thank you again.
Natasa Mehle

Reviewer 2 Report
The manuscript shows valuable optimising of dPCR method for detection and quantification of pepino mosaic virus. I have only few formal reminders to the text.
- Line 38: “…various factors, including genotype, climate and tomato cultivar” – I suggest following specification: “…various factors, including virus genotype, climate and tomato cultivar”
- Lines 47 and 59: Is the appearance of natural genotype-mixed infections not in contradiction with the cross-protection phenomenon? Is there any mention about this fact in the literature?
- Lines 73-75: “…due to the high variability of the PepMV isolates, mismatches in primer or probe sequences can influence this quantification. Moreover, reverse transcription and PCR inhibitors can have large effects on these assays.” – that are general problems of PCR techniques, not especially qPCR. They are presented, however, as disadvantages of qPCR compared to dPCR.
- Line 141: “mix of Ch2, Eur and US1 genotypes” from the Table 1 is not satisfactory explained. Was it prepared by mixing isolated RNAs, isolated viruses, by isolation from mix-infected plants, what was the ratio of particular components…?
- Line 165: In the legend to the Fig. 1 – “…horizontal line (left, 2500; right, 1500)” should be correctly according the Figure: “…horizontal line (left, 3000; right, 1500)”
- Line 349: “…very low fluorescence intensity…” – should be quantitatively specified
Author Response
Dear Reviewer 2,
We would like to thank you for your valuable comments, which helped to improve the manuscript. We have carefully revised the manuscript based on your and other suggestions. Please see below a response to the comments provided by you.
- We added specification as you suggested: “…various factors, including virus genotype, climate and tomato cultivar” (line 39).
- You are right: what is known from mix infections does not supporting using cross-protection. However, to my best knowledge there are no scientifically supported reports which can be let’s say against use of this vaccine. In addition, the vaccine is used in some countries and up to now (at least to my knowledge) without reporting any problems. Of course, cross-protection depends strongly on the genotype, and it is effective only under well-defined conditions. One of these conditions is also accurate determination of the concentration of the PepMV.
- Yes, we agree that mismatches in primer sequences, reverse transcription and PCR inhibitors can have large effects on all PCR techniques, and not only to qPCR. We corrected this sentence in a way that hopefully it is now more clear that it is not just for qPCR (see line 79).
- mix of PepMV-Ch2, PepMV-Eur and PepMV-US1 was prepared from RNA-positive samples. The mix contained the same volume of RNA of each PepMV genotype. More clear explanation is now added below the Table 1.
- The legend to the Figure 1 is corrected in a way that for the left horizontal line value 3000 is written.
- Very low fluorescence intensity mean less than 1000 above the threshold. This is now quantitatively specified (see lines 359 and 388).
Thank you again.
Natasa Mehle
